# Wine economy in Byzantine Shivta (Negev, Israel): Exploring the role of runoff agriculture and droughts through Agent-Based Modeling

Barak Garty[1]*, Gil Gambash[1], Sharona T. Levy[2], Guy Bar-Oz[1]

1 School of Archaeology and Maritime Cultures, University of Haifa, Haifa, Israel, 2 School of Learning Sciences and Leadership in Education, University of Haifa, Haifa, Israel

* bargarty@gmail.com

## Abstract

Viticulture was a vital agricultural and economic activity during the Byzantine period, also in marginal regions like the Negev Desert. Innovative dryland farming techniques, such as runoff harvesting systems, terraces, and pigeon towers, enabled intensive grape cultivation and a thriving wine export economy. This study focuses on the resilience and adaptability of viticulture in the hinterland of Shivta, analyzing how climatic challenges like aridification and drought tested Byzantine water management strategies. The AGENTS model, developed in NetLogo, integrates various components to simulate viticulture dynamics in the Zetan watershed, calculating water availability, crop yields, and labor costs. The results show that higher runoff ratios improve yield efficiency, while excessive runoff ratios diminish productivity. Prolonged droughts significantly decrease wine production and extend recovery times beyond a decade. Wetter climatic scenarios slightly enhance yield efficiency but do not overcome structural limitations, highlighting the fragile nature of viticulture in the Negev desert. Overall, this study highlights the importance of effective water management in sustaining agriculture and the constraints that limited resilience in Shivta's agricultural system. The modeling approach offers insights applicable to other regions and historical contexts facing environmental challenges.

## 1. Introduction

Viticulture was a prominent form of agriculture throughout Mediterranean history and one of the most profitable Mediterranean crops [1–4]. During the Byzantine period (4th to 7th centuries CE in the Southern Levant), the demand for grapes reached its peak. This increase in demand led to the growth of viticulture into the semi-arid and arid regions of the southern Levant, including climatically marginal areas, such as the Negev Desert. It is then, that vineyards provided the foundation for intensive grape cultivation, and wine production facilities dramatically shaped the geographical

**Data availability statement:** All relevant data are within the paper and its Supporting Information files. The AGENTS model code and supporting materials (data files and documentation) are publicly available on GitHub: https://github.com/BarakGarty/AGENTS. The model is also archived with a DOI on COMSES: https://doi.org/10.25937/b356-wh64.

**Funding:** G. B. O.: European Union European Research Council (ERC) Horizon 2020 Research and Innovation Program (Grant No. 101096539). The funders had no role in study design, data collection and analysis, decision to publish, or preparation of the manuscript.

**Competing interests:** The authors have declared that no competing interests exist.

landscape of the Negev. The prosperous intensive viticulture and wine production in the challenging environmental settings of the Negev relied on sophisticated dry-land agriculture, specially adapted to the arid conditions of the Negev desert [5–6].

The extensive archaeological evidence demonstrating relevant dryland agro-technological innovations includes: (1) numerous human-made stone terraces, water channels, dams, and cisterns that were constructed to hold alluvial soil and capture seasonal floodwater to optimize vineyard cultivation in the marginal areas of the Negev [7–8]; (2) pigeon towers built near fields to produce fertilizer to enrich the nutrient-poor desert soil [9–11]; (3) numerous large-scale communal winepresses located within Negev sites [12]; (4) large numbers of wine amphorae and grape seeds found in the refuse dumps of Negev settlements [13–15]; and (5) grapevine pollen discovered in agricultural fields, along with abundant vine twigs found in pigeon towers in the hinterland of Negev sites [16–18].

Various written sources relating to the Negev emphasize the significant role of the wine industry in the region's economy and its global reputation. The Nessana Papyri, dating from the 6th to 7th centuries CE, provide evidence of vineyards located around Nessana [19]. Additionally, other historical texts from the 4th to 6th centuries CE refer to the vineyards of the Negev in relation to winemaking, indicating that grape cultivation was a preferred agricultural practice [1,20]. The robust export market for Negev wine is evidenced by iconographic materials that illustrate the extent of the wine industry in Byzantine Negev. For instance, mosaics discovered in churches in the Negev depict the overland transportation of Gaza jars, which were presumably filled with wine [21]. These Gaza jars were commonly used as transportation vessels for Gaza wine, and have been found in large quantities throughout the Mediterranean world and beyond. Their distribution and abundance in Eastern Mediterranean administrative and religious centers highlight the thriving wine industry that relied on a strong export market, which began in the late 4th to early 5th centuries and peaked in the 6th century before sharply declining starting from the mid-6th century CE [4,14,22–23]. This decline corresponds with the periods of prosperity and downturn in Negev viticulture and wineries [4,14].

The Byzantine viticulture enterprise in the Negev was not just an agricultural venture; it also served as a significant economic driver in a resource-limited region. The labor-intensive nature of dryland viticulture was designed to maximize the harvesting of rainwater runoff, indicating a substantial investment in maintaining this industry. The high demand for wine during the Byzantine period likely motivated these efforts. However, the ability of this industry to withstand climatic and economic challenges remains a critical question.

While the scale of desert runoff agriculture in the Negev desert is agreed to have been significant, the effectiveness of the runoff harvesting infrastructure for enhancing grape production has not been tested. This is especially crucial in the arid Negev region, where climatic stressors, such as drought, are expected to affect the sustainability and productivity of viticulture. This study aims to explore the dynamics of viticulture as both an agricultural and economic venture in the climatically marginal environment of the Byzantine Negev. It seeks to examine how resilient viticulture

was as an economic driver when faced with climatic and environmental challenges. Our case study will be the agricultural landscape around the Byzantine agricultural village of Shivta. To address these archaeological analytical research questions, we employed Agent-Based Modelling (ABM) methodologies to analyze how agriculturalists in the Negev desert managed to cope with the harsh environment.

## 1.1 Location and environment of study area

The Zetan watershed is situated in the northern Negev highlands (Har Hanegev) of Israel, to the east of Mount Boqer. It encompasses the Zetan and Karcha streams, with the latter being a significant tributary to the Zetan, both of which flow into the Lavan stream. Covering an area of 25.08 square kilometers, the Zetan basin is home to the ancient settlement of Shivta, now recognized as a UNESCO heritage site (Fig 1 provides the geographic context of the research area, while Fig 2 presents a higher-resolution view of the Zetan watershed). This area is dotted with numerous archaeological sites around Shivta, including agricultural terraces, remnants of guard towers, farmhouses, and runoff harvesting systems. These features suggest that the region once thrived as an agriculturally-based community, reaching its zenith during the 4th to 6th centuries.

The Negev highlands have long been shaped by climatic fluctuations that impacted settlement and agricultural patterns. While some studies propose a transition from wetter periods before the 1st century CE to a drier climate by the 6th and 7th centuries CE [24–26], the widely accepted view – adopted also by this research – is that the Negev's aridification was a gradual process that began at the end of the first century CE and intensified around the 3rd and 4th centuries, with the Byzantine period largely mirroring present-day arid conditions [27–30].

Given this climatic backdrop, the resilience of Byzantine agricultural systems, particularly in arid regions such as Shivta, offers a unique lens to evaluate how human societies managed climatic stressors. By employing an agent-based model approach with present-day climate data as a proxy, this study investigates the adaptability and resilience of ancient viticulture practices under varying climatic scenarios. While gradual drying trends shaped the region's long-term agricultural challenges, episodic droughts likely exerted acute stress on viticulture, testing the limits of Byzantine water management strategies. This is exemplified in a letter by Procopius of Gaza, who describes drought and wind-driven sandstorms damaging vines and vineyards near Elusa, illustrating how short-term acute climatic stressors could undermine both

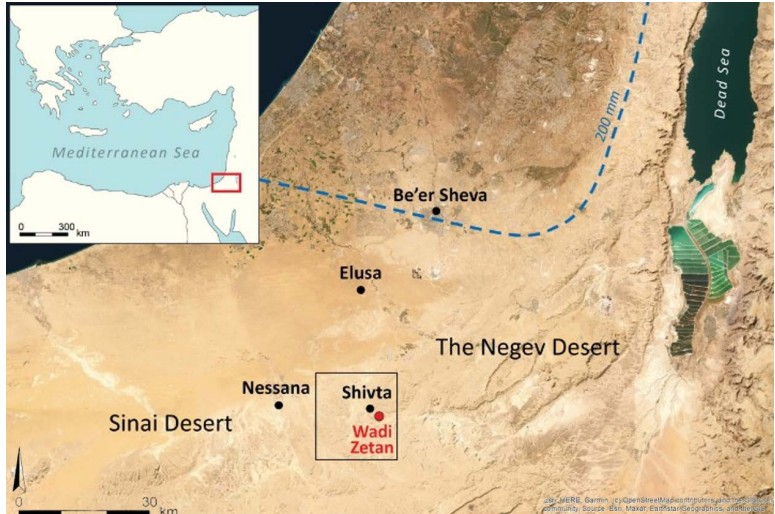

**Fig 1. Geographic context of the research area in the Negev Desert, Israel.** The black square marks the Zetan watershed, the study area around Byzantine Shivta. Basemap source: Esri, Maxar, Earthstar Geographics, and the GIS User Community.

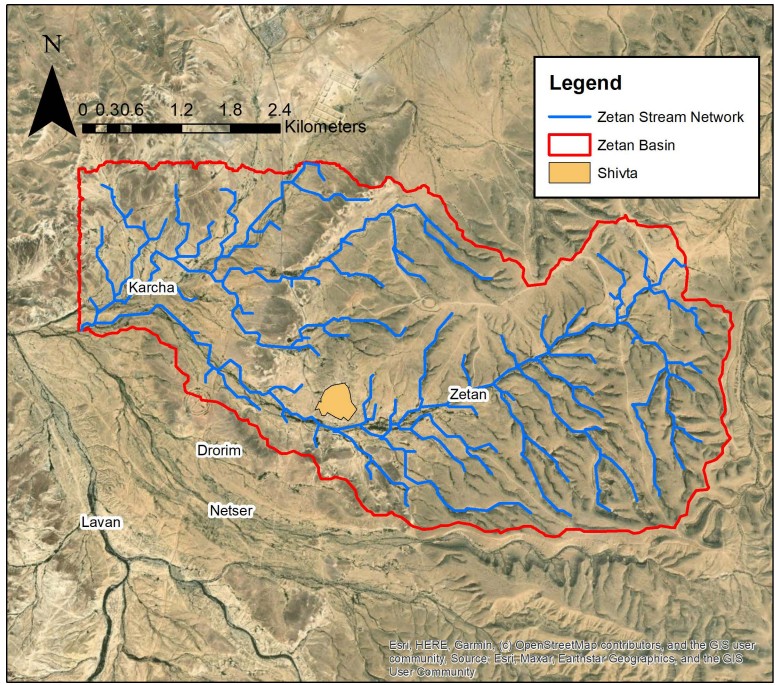

**Fig 2. Research area – Zetan watershed boundary, stream network, and the ancient settlement of Shivta.** Basemap source: Esri, Maxar, Earthstar Geographics, and the GIS User Community.

agricultural viability and broader socioeconomic resilience in the region [31]. Understanding how farmers dealt with such challenges may provide a valuable historical perspective on sustainability and innovation in marginal environments.

While we agree that the Negev's climate during the Byzantine period largely mirrored present-day arid conditions, periods of increased rainfall may have offered temporary relief to water-stressed agricultural systems. Simulating the potential impact of wetter climatic conditions on Shivta's viticulture provides an opportunity to assess whether such shifts could have meaningfully improved agricultural resilience or whether structural limitations imposed inherent constraints on productivity.

## 2. Methods

To evaluate the Byzantine Shivta wine economy under climate stressors, an Agent-Based Model (ABM) was programmed using the NetLogo platform [32] (version 6.4.0) with assistance from the ChatGPT LLM [33]. This model, named AGENTS – Agriculture Grape (yield) Evaluation (using) NetLogo Technology Simulation – integrates environmental, agricultural, and economic data to simulate historical viticulture in the Zetan watershed [34]. The version of the AGENTS model used in this research is available for download at link.

To investigate the impact of climate stressors on wine economy and local viticulture, the methodology included four layers of computation: spatial, climate, agriculture and human behaviors. First, a spatial model representing the Zetan watershed was developed. Next, climate factors such as precipitation and evaporation were incorporated. Then, the agricultural component was integrated, comprising farm units, vineyards, and staple crops like wheat, along with yield calculations and labor costs assessments. Lastly, rules were established, to govern the behavior of farms, including a decision-making mechanism for expansion. Supplement S1 provides a series of instructional videos presenting the model's key functionalities and processes.

The AGENTS model operates on a daily time resolution, where each simulation tick—a fundamental time step in NetLogo—represents a single day, and a full year consists of 365 ticks. Climate-related processes, including precipitation, evaporation, soil moisture, runoff generation, and infiltration, are computed on a daily basis. In contrast, key agricultural and economic decisions—such as yield calculations, farm expansion choices, and memory updates regarding past yields—are executed annually at the end of each 365-tick cycle. This distinction ensures that short-term climatic variability is captured while decision-making processes align with the seasonal nature of historical agricultural practices.

## 2.1 Modeling the spatial layer: elevation, slopes and fluvial features

The NetLogo AGENTS model for the Zetan watershed employs geospatial techniques to simulate interactions between land use, topography, and hydrography. Built on a grid, each patch represents 30 meters by 30 meters section of the terrain, defining the resolution of measurement using high-resolution Digital Elevation Models (DEM) for elevation, slope, and water flow. Patches are categorized into natural terrain, agricultural, and urban areas, each with specific attributes relating to soil, hydrology and land use (i.e., farm, terrace or flood path). This multi-layered structure of computation enables the investigation of historical agricultural practices and urban development, emphasizing their dependence on topography, climate, water resources, and land management strategies (a list of the main data sources and key publications the model was based on is provided in S2). By integrating GIS data on stream networks and land use patterns, the model provides insights into the environmental dynamics of the ancient Shivta and Zetan regions. Fig 3 illustrates the data preparation actions taken with the ArcGIS Map before importing it into NetLogo.

## 2.2 Modeling climate layer: precipitation, evaporation and hydrology

To simulate ancient grapevines and wheat cultivation the model calculates the water balance for agricultural terraces, determining water availability during each growing season to estimate potential crop yields by weight per unit area. The water balance equation, which normally includes factors like precipitation, runoff, and evaporation, was simplified for ease of use in its final form. In the model, these components were accounted for as concurrent processes at a daily resolution: direct rainfall, harvested runoff, and evaporation were calculated at the terrace level, with excess water redistributed downslope. Additionally, infiltration was subtracted daily from the terrace water column, representing the amount reaching the crop. The total infiltrated water over a full year was then stored in the model variable available water storage and used in the final yield equation as a single aggregated value of seasonal water availability. This decision is supported by findings that rainfall is the most significant weather component influencing yield, accounting for 50–60% of variability in grapevines [35–36] and up to 80% in wheat [37]. Similarly, Stavi et al. [38] highlight that in terrace agriculture, water availability is primarily governed by infiltration dynamics, which reflect the balance between precipitation, runoff, and evapotranspiration.

Incorporating current precipitation data into the AGENTS model requires evaluating changes in the Negev's climate since Shivta's demographic peak, 1500 years ago. The annual precipitation data collected during experiments in Shivta in the 1960s [5] proved insufficient for the model's requirements. To address this issue, long-term daily rainfall data from Sede Boqer (1952–2022) and Shivta (1963–1971) were acquired from the Israel Meteorological Services (IMS) [39]. Using the approach suggested by Dafni [40], a strong correlation between the rainfall datasets from these two locations was found (R²=0.83). This allowed the reconstruction of the missing years for Shivta using data from Sede Boqer. The reconstructed dataset was validated by comparing annual averages, revealing a minimal deviation (3.7%) between the original and reconstructed data (S3). Evaporation data for Byzantine Shivta, which is crucial for calculating water availability in agricultural terraces, was not available directly. To overcome this challenge, we adapted evaporation data from nearby IMS stations [39] using a method that correlates mean daily temperature with evaporation, based on findings by Ansorge and Beran [41]. This approach enables to estimate daily evaporation rates with strong statistical reliability. First, we confirmed that temperature data from Be'er Sheva IMS station [39], located 40 km northeast of Shivta, could be used

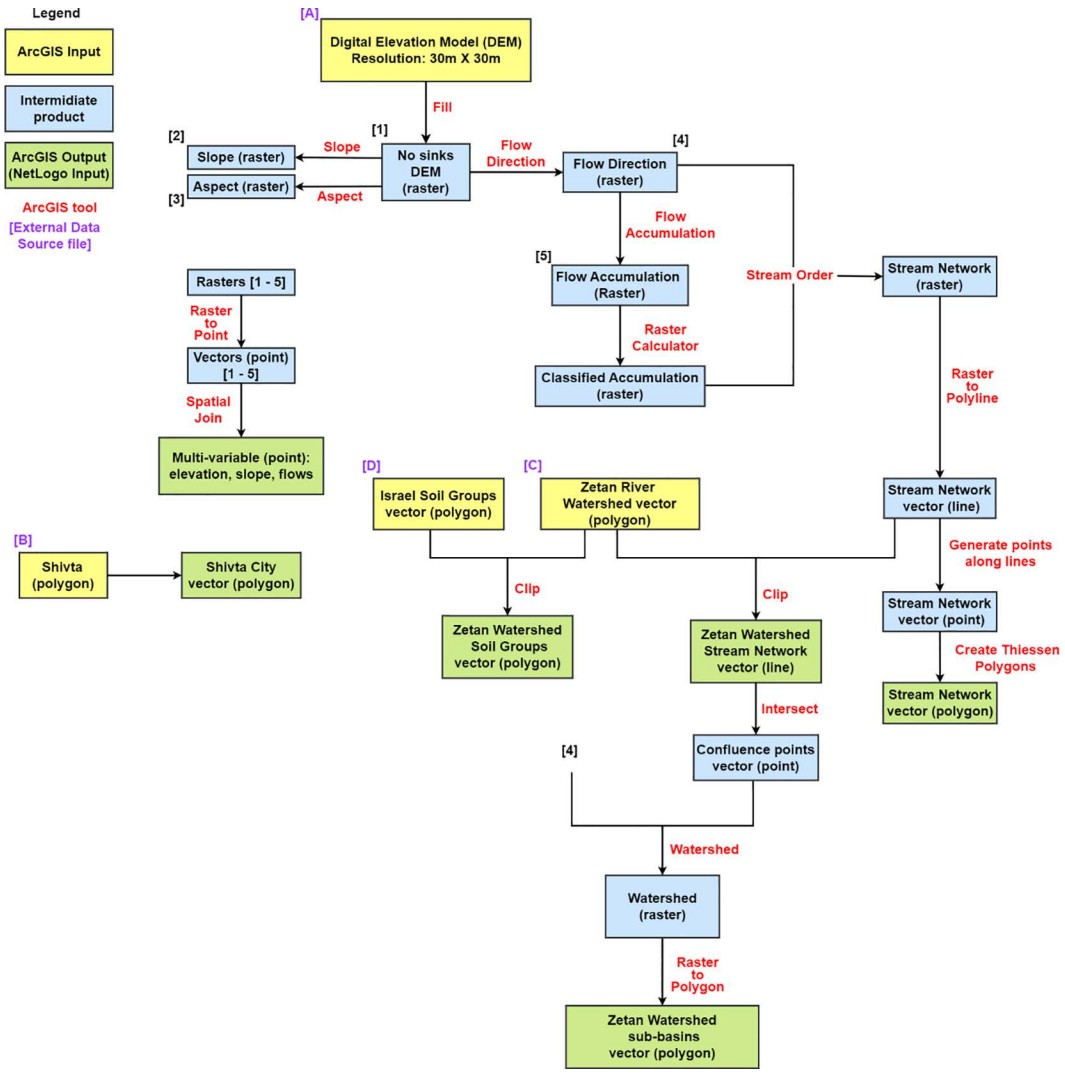

**Fig 3. ArcGIS products preparation for the NetLogo AGENTS model workflow diagram.**

as a proxy for Shivta. This was verified by comparing daily mean temperatures from both locations (1965–1971), which showed high correlation (R²＝0.98), indicating similar temperature regimes. Subsequently, we developed an equation to estimate daily evaporation for Be'er Sheva based on its mean daily temperatures (R²＝0.73). This allowed us to convert the mean daily temperature dataset from Be'er Sheva into evaporation data, which could be used in the model to simulate the missing evaporation data for Shivta (S4).

Runoff estimation in the AGENTS model employs the Soil Conservation Service Curve Number (SCS-CN) method, a recognized approach for calculating runoff potential [42]. This method is suitable for the semi-arid Negev Highlands due to its simplicity and adaptability to varying environmental conditions. The SCS-CN method uses curve numbers (CN) to estimate runoff based on soil type, land use, and moisture conditions, with values adjusted for local factors like soil textures and slopes in the Zetan catchment area [43–46].

Daily precipitation data drives the model's runoff calculations, with CN values assigned based on its hydrological soil group, texture, and vegetation. The values used for this study, detailed in Table 1, were based on previous research on

**Table 1. Zetan watershed soils classification, textures and Curve Numbers values.**

| Soil main group index [51] | Soil sub-group index [52] | Soil texture [53] | Hydrologic soil Group [54] | AGENTS model CN index value |
|---|---|---|---|---|
| R: Loessial Serozems | R1: Loess of the Negev Lowlands and the Negev Mountains – the Sirozium area | silt loam (very fine) | D | 92 |
| | R4: Coarse desert loess and alluvium | | | |
| S: Brown Lithosols and Loessial Serozems | S1: Exposed rocks and loess | Loam | C | 85 |
| | S2: Brown lithosol and loess | | | |
| | S3: Sirosium loess and brown lithosol | | | |
| X: Bare Rocks and Desert Lithosols | X6: Reg Lithosol and loess | Rocks and gravel with little sediment (Loam) | B | 77 |

Shivta's soils [5,47] and aligned with other studies in Israel [48–50]. Adjustments accounted for desert soil characteristics and steep slopes (for complete equations and variables see S5).

The runoff ratio, which relates the runoff-producing catchment area to the cultivated receiving area, is derived from studies of ancient runoff farms, particularly in the Negev Highland, where it typically ranges from 1:9–1:33, averaging around 1:20 [5,55]. Ancient farmers improved runoff collection by clearing slopes, resulting in the formation of stone mounds, which enhanced runoff efficiency by approximately 30% due to the clay properties of Negev soils [5,48]. In the AGENTS model, this increased efficiency is represented by multiplying runoff amounts by 1.3 for 'clear with mounds' patches. The model captures the interaction between climate, hydrology and agriculture through daily water balance simulations influenced by precipitation and evaporation. Each terrace, based on a 30x30 meter digital elevation model (DEM), accumulates water as an above-ground column with a maximum storage capacity. Excess water flows to adjacent patches based on topography, while infiltrated water is used to calculate crop yields.

The model's runoff mechanism and Curve Number (CN) values were calibrated using data from 239 runoff events at the Nitzana hydrometric station from 2009–2023 [56]. The Zetan basin contributed 12% to the Nitzana watershed, allowing for scaling of runoff data. Simulations indicated the reported runoff was 56% greater than initial predictions, resulting in calibration factor of 1.567 to accurately reflects observed runoff volumes (calibration process and validation are available in S6).

### 2.3 Modeling agriculture layer: farms, crops, yield and labor

The AGENTS model simulates Byzantine agricultural practices in Shivta through three components: farms, terraces, and crop yields. Farms represent households managing terraces for staple wheat fields and cash crop or grapevine, reflecting the region's historical reliance [57].

Annual Crop yields depend on water availability, with wheat yield equations based on studies from the Negev [58–60]. These works, combined with findings from the Avdat farm experiment [5], establish a relationship between yield and precipitation, adjusted for the AGENTS model's terrace size (0.09 hectares). For grapevines, yield equations used comparative data from Castilla La Mancha, Spain, where rainfed viticulture [61] correlates well with precipitation [62]. The model also incorporates qualitative thresholds from Jackson and Schuster [63] and Kedar [64] to account for climatic constraints and wine quality factors. A 10% reduction in yields reflected ancient pest control inefficiencies compared to modern practices [65–66]. See S7 for yield plots and equations.

Labor costs are crucial, covering energy and resource for terrace construction and crop-specific operations. These are measured in man-days converted to wheat-equivalent costs using FAO standards [67,68]. Detailed labor costs are provided in Table 2, and the calculations underlying these values can be found in S8 [69–72].

At each time-step, the farm evaluates its wheat storage against labor costs needed for maintenance and new constructions at the beginning of each season. If the farm's storage is insufficient, certain fields and terraces are marked inactive, and supply no yield in the following season. This cyclical process dynamically reflects resource availability and productivity.

The model also tracks terrace-level statistics, including Total Yield, Average Yield and Yield Efficiency, updated each season for both wheat and grapevines (see S9 for model and research glossary). Rigorous evaluations ensured that yield output aligned with historical benchmarks. For grapevines, simulated yields averaged 339 kg per terrace (0.09 hectares), consistent with historical data, including Columella's yield ratios (*Rust*. 3.3.10) and Jongman [74]. Similarly, wheat yields averaged 38.3 kg per terrace, aligning with Columella's threefold yield-to-sow ratio (*Rust*. 1.3.4). No calibration adjustments were necessary (see S10 for detailed analyses).

### 2.4 Modeling human behavior: governing rules and farmers' decision-making

It is assumed that grapevine cultivation and wine production in Shivta were economically motivated cash crops, reflecting historical accounts of their profit value (Cato *Agr*. 1.7; Columella *Rust*. 3.3). Economic decision-making in the model draws on rational choice theory, which posits that individuals make optimal decisions under constraints [75], and bounded rationality theory, which acknowledges that limited information can lead to suboptimal choices [76].

Bernoulli's [77] Expected Utility Theory highlights that options are prioritized based on preferences, while Kahneman and Tversky's [78] Prospect Theory introduced psychological factors like loss aversion and the framing effect, emphasizing that losses are perceived more strongly than equivalent gains.

The AGENTS model Implements loss aversion through a memory variable tracking past seasonal production for wheat and grapevines. Farms categorize yields as success or failure based on established thresholds: for grapevines, yields below 270 kg per terrace are failures while above 400 kg are success. For wheat, yields under 10 kg are failure, and above 20 kg are success (Columella, *Rust*. 1.3.4; Table 2 wheat related entries; S10).

At the end of each season, farms update their average yields and memory records, which inform decisions for the next season. High grapevine yields prompt vineyard expansion, while moderate yields depend on the farm's history of failures. Low yields lead to reassessment of wheat fields. This process fosters adaptive behavior based on past

**Table 2. Farm labor costs per 0.1 hectares in kg of wheat.**

| Activity[a] | Man- days[b] | Wheat (kg)[b] | Source |
|---|---|---|---|
| Building a stone terrace | 100 | 90 | [64: 68] |
| Creating wheat field | 4 | 3.6 | [5: 179–190, 69: 284–285] |
| Clearing a runoff field | 20–30 | 18–27 | [5: 132] |
| Wheat field season agriculture activities[c] | 8 | 7.2 | Columella (*Rust*. 2.12.1) [70: 329, 71: 138, 72: 68] |
| Wheat sown | -- | 14.8 | Varro (*Rust*. 1.44.1) Pliny (*HN* 18.12) |
| Vineyard: soil preparation, digging holes and planting vines | 27.8–46.3[d] | 25.0–41.6 | [70: 331] |
| Vineyard: age 1–3 maintenance (excluding harvest)[e] | (11.4–15) X3 | (9.2–12.2) X3 | [70: 331, 71: 136] |
| Vineyard: age 4+ maintenance (including harvest) | 14.2–17.8 | 12.8–16 | [70: 331, 71: 136] |

[a]Each year an additional 10% maintenance costs for the infrastructure are forced on each farm [73].

[b]When a range of values was available, the model used the mid-point value of the range given.

[c]Wheat growing is done in fallow pattern, alternate years, in model level this was coded as only half a wheat filed output is calculated and thus only half the costs are required.

[d]Labor based on modern planting density, following rainfed vineyard standards of 3 m between rows and 1.5 m between vines—yielding 2,220 vines/ha or 222 vines per 0.1 ha—consistent with the model's yield equation.

[e]First 3 years of vineyard accounted as single cost.

results, making farms increasingly risk-averse after consecutive failures and driving the simulation's agricultural strategies.

In summary, the AGENTS model integrates layers of spatial, climate, agricultural, and human behavior to simulate the socio-economic and environmental dynamics of Byzantine viticulture in the Zetan watershed. By combining historical data, GIS techniques, and validated computational methodologies, the model provides a comprehensive framework for evaluating the interplay between climate stressors, agricultural practices, and economic decision-making. This framework not only advances our understanding of ancient farming systems but also offers a flexible tool for testing various scenarios, contributing to broader discussions on resilience and sustainability in arid-zone agriculture.

## 2.5 Exploring the model: climate stressors affecting Byzantine Shivta wine production

To evaluate the resilience and sustainability of viticulture in Byzantine Shivta under varying climatic stressors, the AGENTS model underwent targeted tests using the NetLogo BehaviorSpace application, which supports systematic variation of multiple variables and repetitions (see S11 for details of the BehaviorSpace configurations).

**Test 1: Runoff ratio and yield efficiency** – This test assessed vineyard yield efficiency based on runoff ratios ranging from 1:5–1:30. 40 simulations were run for each ratio over a decade of randomly selected precipitation years, quantifying the relationship between water availability and vineyard productivity.

**Test 2: Impact of consecutive droughts** – The effect of varying drought durations on wine production was explored by simulating a decade with 0–5 consecutive drought years (defined as less than 66 mm rainfall: [79]). Each drought scenario underwent 100 model runs to reveal how drought duration correlates with average wine production.

**Test 3: Recovery time after droughts** – Recovery time for wine production post-drought was measured over 20-year periods, starting with a 5-year non-drought period followed by droughts of varying length. Recovery time was assessed based on the years needed to surpass pre-drought production levels, with each drought length tested 100 times.

**Test 4: Wetter climate scenarios** – Simulations under wetter scenarios (10% and 25% more rainfall) examined how increased precipitation impacts vineyard yield efficiency, analyzed across the runoff ratios from Test 1.

The AGENTS model effectively evaluates the relationship between climate variables and agricultural performance in Byzantine Shivta. It highlights the effect of drought, recovery potential, and the advantages of increased rainfall, shedding light on the adaptive strategies that influenced viticulture in the semi-arid Negev.

## 3. Results

To evaluate the key factors influencing viticulture yields in the semi-arid Byzantine Negev Shivta and to assess the sustainability of its grapevine agriculture under harsh, arid conditions, four experimental simulations were conducted. These simulations examined the principal factors affecting vineyard yield efficiency and overall wine production, considering the limited resources of the area.

Fig 4 compares the modeled agricultural system of ancient Shivta in the Zetan watershed with archaeological data by overlaying the model's outputs onto a GIS layer derived from documented agricultural features found in excavations (see data in [80]). The model places grapevine terraces and wheat fields near flood-prone areas, often overlapping with documented agricultural zones. Additionally, the simulated locations of stone mound areas and farmhouses align well with published archaeological data, further validating the model's accuracy.

Additional details on the model test setups and analyses are available in the supplemental files accompanying this article. S12 provides the yield efficiency versus runoff ratio model test setup and a detailed SPSS analysis of these results. S13 contains the model setup and SPSS analysis focusing on the impact of drought years on the capacity of the Shivta wine industry. S14 includes the model setup and an in-depth SPSS analysis on the recovery time of the Shivta wine industry in relation to varying drought durations. Finally, S15 presents the setup and SPSS analysis for the effects of wetter climate conditions on the modeled yield efficiency of grapevines in Shivta.

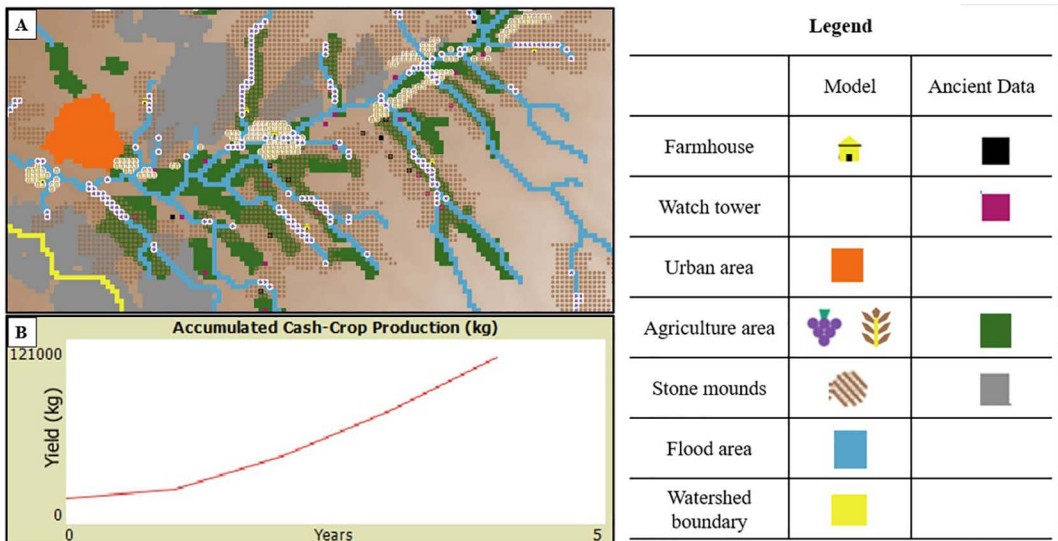

**Fig 4. AGENTS model simulation output.** [A] Model agriculture system (against Shivta archeological reported agriculture system (after [80]). [B] Accumulated 5-years grapevine yield (kg) from the entire modeled agriculture system.

## 3.1 Yield efficiency as a function of runoff ratio

The success of viticulture in Shivta's arid climate depended heavily on the rainwater runoff captured from the surrounding hills to the vineyards constructed in riverbeds. Comparing yield efficiency to runoff ratio shows that maximum overall efficiency is achieved when the runoff ratio is 30 units of runoff area per unit of vineyard (see Table 3 for test statistics and Fig 5 for plotted results).

As expected, increasing the runoff ratio maximizes the capturing of rainwater runoff in the vineyards, thus leading to increased yields. This linear relationship translates directly to higher total grape yield and, consequently, to more wine production per terrace unit.

However, this linear relationship does not hold for all runoff ratios. At a certain point, according to the grapevine yield function (S7), too much water can cause a decline in yield. We did not test the model with runoff exceeding a 1:30 ratio, as this is the maximum recorded in archaeological surveys from northern Negev Byzantine settlements, particularly in Shivta Evenari et al. [5].

## 3.2 Impact of consecutive drought years on Shivta's wine industry

The results show an increasing impact of drought length on the annual wine production of the Shivta Zetan watershed vineyards, with longer drought periods significantly reducing output compared to a decade without drought stressors (see Table 4 for test statistics and Fig 6 for plotted results).

**Table 3. Descriptive statistics of Yield efficiency as function of runoff ratio.**

| Variable Name | N[1] | Min | Max | Mean | S.D. |
|---|---|---|---|---|---|
| Runoff Ratio | 26 | 5 | 30 | | |
| Grapevine Yield Efficiency | 26 | 0.242 | 0.375 | 0.316 | 0.037 |

[1]Based on 1,040 random decades scenarios.

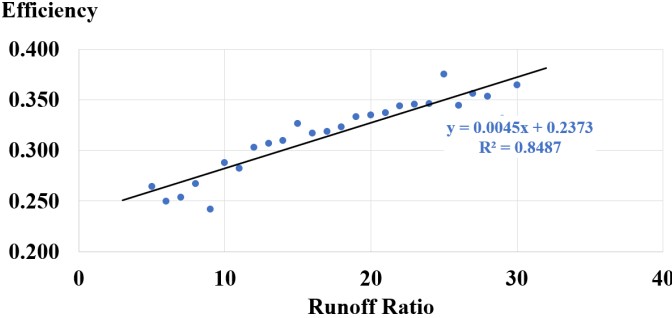

**Fig 5. Grapevines terraces yield efficiency as function of runoff ratio.**

**Table 4. Descriptive statistics of droughts length and mean wine production (liters x1,000).**

| Drought Length | N | Mean | S.D. | 95% Confidence Interval for Mean | | Min | Max |
|---|---|---|---|---|---|---|---|
| | | | | Lower Bound | Upper Bound | | |
| None (0 years) | 100 | 4.41 | 1.75 | 4.06 | 4.75 | 1.17 | 9.42 |
| Short (1–2 years) | 200 | 3.19 | 1.38 | 3.00 | 3.38 | 0.56 | 8.72 |
| Medium (3–4 years) | 200 | 2.17 | 0.98 | 2.03 | 2.30 | 0.24 | 7.30 |
| Long (5 years) | 100 | 1.53 | 0.69 | 1.40 | 1.67 | 0.22 | 4.77 |
| Total | 600 | 2.77 | 1.56 | 2.65 | 2.90 | 0.22 | 9.42 |

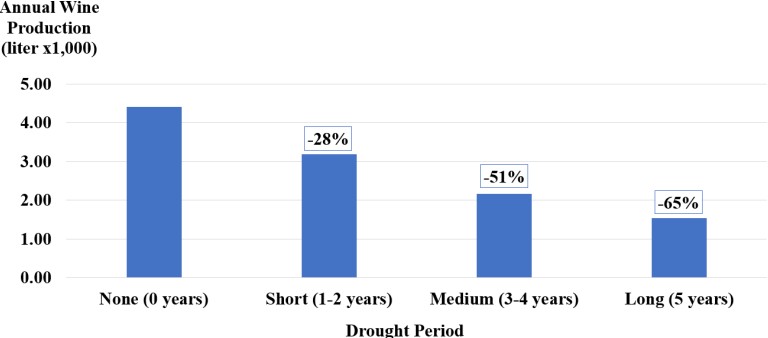

**Fig 6. Mean wine production (liters x1,000) and overall productivity reduction (%) across different drought lengths (N=600).**

A short drought period of 1–2 consecutive years, was found to result in a 28% reduction in average annual wine production over a 10-year period compared to an equivalent no-drought decade. If the drought persisted for 3–4 years, the reduction increased to over 50%. An overwhelming 65% reduction in productivity was observed when the drought extended to 5 consecutive years.

### 3.3 Shivta's wine industry recovery time

Recovery time of Shivta's Zetan agriculture system in terms of overall wine production (liters) test results indicate that after a 2-years drought period, the agricultural system takes an average of 3.4 years to regain its pre-drought production capacity, with 5% of all scenarios failing to recover even after a decade following the drought's end. A medium 3–4-year

drought requires approximately 4.9 years for recovery, with a 21% chance that the system will not return to pre-drought levels even 10 years after the drought ends. A prolonged 5-year drought significantly impacts the wine industry of Shivta, leading to a recovery time of nearly 6.6 years, with almost a 40% chance of failing to recover. Refer to Table 5 for test statistics and Fig 7 for plotted results.

### 3.4 Impact of wetter climates on grapevine yield efficiency as a function of runoff ratio

Testing for wetter climates of +10% annual precipitation (from Shivta's annual average of 87 mm; IMS [39]) and +25% annual precipitation with runoff ratio spreading from 1:5–1:30 (similar to test results presented in section 3.1) shows that grapevine yield efficiency increases as annual precipitation increases. Refer to Table 6 for test statistics and Fig 8 for plotted results.

Model results indicate a 9.0% increase in yield efficiency with a 10% rise in annual precipitation, equating to 200 kg/0.1 ha according to the yield formula (S7, S15). With a 25% increase in annual precipitation, yield efficiency could improve by 17.4% over normal conditions, reaching approximately 280 kg/0.1 ha.

## 4. Discussion

The AGENTS model provides new insights into the resilience of Byzantine viticulture in Shivta to climatic stressors. By simulating historical runoff harvesting and terrace cultivation practices, the model demonstrates the capacity of these techniques to buffer vineyards against variability in precipitation. Results highlight that while moderate drought conditions could be mitigated through efficient water management, prolonged dry periods posed significant challenges to maintaining agricultural productivity.

While the model's yield estimates are based on modern rainfed viticulture data from Spain, important differences must be acknowledged. Climatic conditions, soil composition, agricultural practices, and grape cultivars in the Byzantine Negev likely

**Table 5. Descriptive statistics of regain productivity period (years) and drought length.**

| Drought Length | N | Mean | S.D. | 95% Confidence Interval for Mean | | Min | Max |
|---|---|---|---|---|---|---|---|
| | | | | Lower Bound | Upper Bound | | |
| Short (2-years) | 100 | 3.37 | 3.39 | 2.70 | 4.04 | 1.0 | 14.0 |
| Medium (3–4 years) | 200 | 4.90 | 4.24 | 4.30 | 5.49 | 1.0 | 13.0 |
| Long (5-years) | 100 | 6.62 | 4.08 | 5.81 | 7.43 | 1.0 | 11.0 |
| Total | 400 | 4.95 | 4.16 | 4.54 | 5.35 | 1.0 | 14.0 |

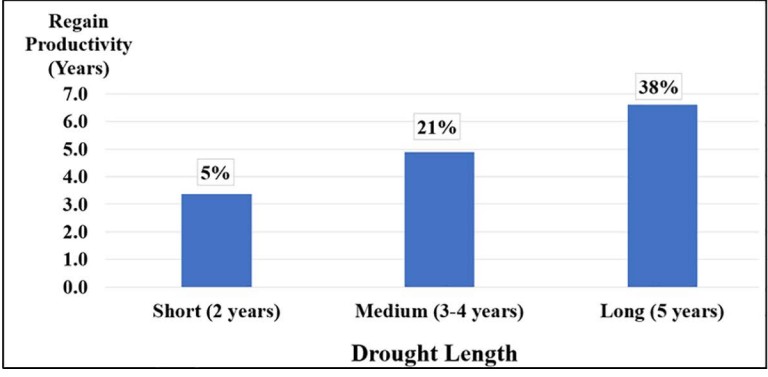

**Fig 7. Impact of Drought Length on Recovery Time (in years) and Probability of Long-Term Productivity Loss in the Shivta Vineyard System (N = 400).**

**Table 6. Descriptive statistics of grapevine yield efficiency as function of wetter climates categories test.**

| Climate Group | N | Mean | S.D. | Min | Max |
|---|---|---|---|---|---|
| same as present-day | 1040 | 0.322 | 0.082 | 0.097 | 0.667 |
| +10% precipitation | 1040 | 0.351 | 0.078 | 0.139 | 0.602 |
| +25% precipitation | 1040 | 0.378 | 0.078 | 0.079 | 0.645 |
| Total | 3120 | 0.350 | 0.083 | 0.079 | 0.667 |

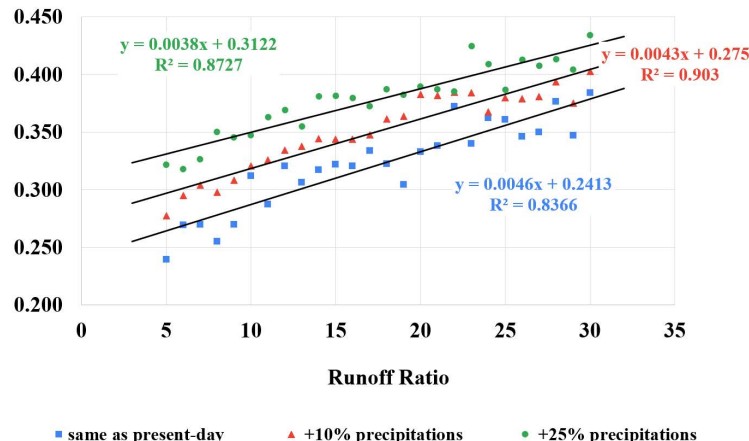

**Fig 8. Grapevine yield efficiency as a function of runoff ratio under different climate scenarios (N = 26).**

differed significantly from those in modern Spain. Therefore, the results should be interpreted as indicative of general patterns in rainfed viticulture under arid conditions, rather than as precise historical reconstructions. The conclusions drawn here aim to explore potential systemic behaviors rather than establish direct equivalence between the ancient and modern contexts.

This study underscores the role of adaptive agricultural strategies in sustaining viticulture in marginal environments where even small variations that are sustained can instigate permanent changes to the system's behavior. The ability to test drought scenarios of varying lengths, starting at different points within a decade, offers a novel perspective on ancient farmers' responses to climatic pressures. Additionally, the ability to simulate daily life and societal dynamics of that period is an innovative aspect of this approach. It enables pinpointing the effects of climate stressors with single-year or decade-level resolution, a granularity previously unavailable to archaeologists, who often analyzed the Byzantine wine industry in the southern Levant over large time spans, such as entire centuries or the Byzantine period (4th to 6th centuries CE).

## 4.1 Yield efficiency and runoff ratio

The relationship between runoff ratio and yield efficiency underscores the critical role of water management in sustaining grapevine agriculture in the Negev. Another important factor likely contributing to vineyard productivity was the enrichment of soils with pigeon droppings. Archaeological studies from Shivta and Sa'adon indicate that dovecotes were deliberately maintained to collect manure, which was then spread on surrounding agricultural fields [9,11,16,81]. Pigeon manure, rich in nitrogen and phosphorus, would have improved fertility in the nutrient-poor desert soils, complementing water management efforts and enhancing crops yields [82,83]. While increasing the runoff ratio enhances the capture of rainwater and improves yield efficiency, there are diminishing returns beyond a certain threshold. This is consistent with archaeological

records indicating a maximum runoff ratio of 1:30 in Byzantine Shivta [5]. The results suggest that farmers were likely aware of the limits of water management techniques, optimizing their runoff areas within practical and economic constraints. However, the labor-intensive construction of runoff harvesting areas and terraces imposes significant costs on the system, limiting the ability of farmers to expand their overall cultivated areas.

If we are to consider the yield efficiency of a modern rainfed grapevine agriculture system such as that of Castilla La Mancha in Spain—whose data informed the AGENTS model grapevine yield formula—their recorded dataset shows yield efficiencies ranging from 67% during drought years (~70% of annual precipitation, e.g., 2017; [61]; yield efficiency adjusted for lack of pest control) to 94%, with an average yield efficiency of 79%. This comparison highlights the stark contrast with Shivta's viticulture, where the yield efficiency of a given vineyard area averages around 31%. Such low efficiency underscores the fragile nature of this agricultural system. Despite considerable effort in harvesting runoff water to cultivate water-demanding crops like grapevines, Shivta's lower yield efficiency can be attributed to its lower overall precipitation, higher drought susceptibility, and the absence of modern agricultural inputs such as fertilizers and pest control, which enhance productivity in contemporary rainfed viticulture.

If we take the model's average yield efficiency of 31% and translate it into actual wine production per 0.1 hectares using the yield formula and a conversion ratio of 0.6 to 0.7 liters of wine per kilogram of grapes [84,85], the results amount to approximately 10–12 hectolitres per hectare. This figure represents only half the production recorded in 17th-century Italian vineyards, which were both rainfed and devoid of modern pest control practices [86]. The inclusion of such historical data underscores the extreme limitations of Byzantine viticulture in Shivta. Despite the Northern Negev Byzantine farmers' considerable efforts to optimize water harvesting and land use, the resulting yield efficiencies remained far below those of other rainfed grapevine systems, even in pre-modern contexts.

Even with these challenges, the significant demand for wine during the Byzantine period may have served as a strong economic incentive to sustain such a labor-intensive agricultural system. While scholars differ on the extent of this demand, with some suggesting it was mainly for local consumption [87,88] and others viewing it as part of a large-scale industry supplying prestigious markets [14], the existence of demand is undisputed. This potential for high market value likely justified the intensive water and labor inputs required for viticulture in such a marginal environment—particularly if the wine was a boutique product of exceptional quality. Such a reputation is supported by the widespread distribution of amphoras associated with South-Palestinian wines across the Mediterranean and beyond, reflecting the region's role in an extensive trade network [4,23].

While higher runoff ratios improve yield efficiency, they also increase production costs. Constructing and maintaining runoff harvesting infrastructure and terraces required substantial labor and materials, which could limit farmers' ability to expand their agricultural operations. Farmers with limited resources may have been forced to prioritize the most productive areas, potentially leaving less efficient plots uncultivated. This economic limitation could explain the observed discrepancies in overall wine production despite optimized runoff ratios. It also highlights the socio-economic stratification that may have existed, with wealthier farmers better able to invest in water management infrastructure and sustain higher production levels. Together, these findings emphasize the interconnectedness of water management strategies and economic constraints in shaping the viability of viticulture in the Byzantine Negev.

## 4.2 Vulnerability to drought

The results highlight the pronounced sensitivity of Shivta's viticulture to drought events. Even short droughts of 1–2 years resulted in a 28% reduction in wine production over a decade, while prolonged droughts caused disastrous declines, with reductions exceeding 65% for 5-year droughts. These findings align with historical climate variability patterns in the Negev, where consecutive dry years were not uncommon [27,89,90]. Such droughts include both climatic droughts, defined as a 25% reduction in annual average precipitation, and hydrometric or agricultural droughts caused by poor rainfall distribution during the wet season [79]. Based on the IMS precipitation dataset [39] used for the AGENTS model simulations, a single

dry year occurs approximately every 2.7 years, while a 2-year drought period happens once every 11.7 years. A 3-year consecutive drought is rarer, occurring once in 50 years [91]. Longer droughts, such as 5 or more consecutive dry years, are extremely rare, occurring only once in a century [79].

The extended recovery periods following droughts—up to 6.6 years post a 5-year drought—demonstrate the fragility of Shivta's viticulture, with prolonged reductions in grape yields severely disrupting wine production and economic stability. This inability to rebound quickly may have led to cascading effects, forcing farmers to abandon their land temporarily or even permanently.

The deliberate sealing of residential buildings at Shivta, documented by Tepper et al. [92], reflects the settlement's decline during the mid-6th century [10,13,93] This abandonment likely stemmed from cascading economic pressures tied to the sharp decline in viticulture, a primary driver of the local economy [14]. The mid-6th-century reduction in grape pip ratios and Gaza amphorae further indicates a steep downturn in wine production [4]. With demand for Gaza and Ashkelon wines plummeting due to Mediterranean market contractions [2], local farmers faced an untenable situation.

The sealed doors, along with archaeological, zooarchaeological, and palaeobotanical evidence of declining agricultural activity [17,94–95], point to a broader agricultural and societal collapse. While some inhabitants left to seek opportunities in urban centers, others likely intended to return once economic or environmental conditions improved, as indicated by the deliberate preservation of their homes. This phenomenon underscores the fragile interconnectedness of viticulture, settlement sustainability, and external demand during the Byzantine period.

## 4.3  Impact of wetter climates

Model tests simulating wetter climates reveal a mild improvement in yield efficiency with increased precipitation. A 10% increase in annual rainfall led to an 11% gain in yield efficiency, while a 25% increase resulted in a 20% improvement. While these increases provided a modest boost to grapevine productivity, they were insufficient to significantly enhance the resilience of Shivta's viticulture. Even under wetter conditions, the overall yield efficiency remained far below that reported for mid-17th century Italy and modern rainfed systems such as 21st-century Castilla La Mancha in Spain.

The persistence of relatively low efficiency highlights the inherent limitations of Shivta's agricultural system. Although wetter climates may have temporarily alleviated water stress, they did not overcome the structural constraints of resource availability, labor demands, and technological limitations. This suggests that even with favorable climatic conditions, Byzantine farmers in Shivta faced substantial barriers to achieving productivity levels comparable to other historical and modern rainfed viticulture systems.

## 5.  Conclusions

In the year 395 CE, we hear from Mark the Deacon (*Vita Porph*. 19–20) that severe drought had occurred in the city of Gaza and its hinterland, presumably including also the Negev Highlands. As prayers and ceremonies dedicated to Gaza's Lord of Showers, the God Marnas, did not help, the local population turned to Porphyry, the Christian bishop representing the new religion, then struggling to gain a foothold in the pagan Southern Levant. It was only the involvement of Jesus Christ, says Mark, that brought the first rain of the year to the locals' relief, as late as in the month of January 396.

While Christianity would have penetrated the Negev sooner or later, the choice to involve Jesus' first appearance there as the new Lord of Showers is telling. This study explored the resilience of Byzantine viticulture in Shivta, a climatically marginal environment, using an agent-based modeling (ABM) approach. The results reveal the critical role of runoff agriculture in sustaining grapevine yields under arid conditions, with maximum yield efficiency achieved at a runoff ratio of 1:30. However, the system's fragility is evident, as consecutive droughts caused significant reductions in wine production, with prolonged recovery periods exceeding six years for 5-year droughts.

These findings underscore the ingenuity of Byzantine farmers in adapting to challenging environmental conditions through advanced water management strategies. Yet, the economic sustainability of this system was constrained by its

high labor demands and vulnerability to climatic variability. The resilience of the Shivta agricultural economy was likely driven by the high demand for wine during the Byzantine period, reinforcing the role of economic incentives in marginal agricultural systems.

As with other ancient societies facing environmental and economic pressures, the challenges and strategies of Byzantine farmers in Shivta offer valuable lessons on the interplay between resource management, economic drivers, and environmental sustainability. This research contributes to broader discussions on the resilience of agricultural systems in marginal environments, past and present, and highlights the potential of ABM methodologies in addressing historical questions.

Future work with the AGENTS model should focus on expanding its applicability and refining its parameters. A generic version of the model could be developed to investigate similar questions in other regions and periods, allowing for broader research applications, such as food security under the resource constraints of ancient societies (see the work of Campmany Jiménez et al. [96]). Comparative analyses between simulation outputs and archaeological records, such as cultivated areas or settlement patterns, would help validate the model (similar to the Anasazi Long House Valley model; [97–98]). Future iterations should also incorporate dynamic social patterns, constraints on agricultural labor, and non-rational decision-making factors to better capture the complexity of historical agricultural systems. Such developments would enable a more nuanced understanding of viticulture and its economic, social, and political dimensions in the Byzantine Negev and beyond.

## Supporting information

**S1. Supplementary materials for AGENTS model simulations and analysis.** This section contains all supplementary materials mentioned in the manuscript (S1 to S15). Maps throughout this article were created using ArcMap® (version 10.8.2) software by Esri, incorporating the World Imagery basemap [99].
(DOCX)

## Acknowledgments

The authors wish to thank Yoav Avni, Gideon Avni, Yoav Borstein, Hendrick Bruins, Daniel Fuks, Noam Greenberg, Shauly Melitz, Efrat Morin, Jon Seligman, Moti Zohar for helping us with various aspects of this project and for contributing their knowledge and time.

This article is dedicated to the memory of Captain Avraham Albert Garty, a cherished family man and source of inspiration.

## Author contributions

**Conceptualization:** Barak Garty, Gil Gambash, Sharona T. Levy, Guy Bar-Oz.

**Data curation:** Sharona T. Levy.

**Formal analysis:** Barak Garty.

**Funding acquisition:** Guy Bar-Oz.

**Investigation:** Barak Garty.

**Methodology:** Barak Garty.

**Software:** Barak Garty.

**Supervision:** Gil Gambash, Sharona T. Levy, Guy Bar-Oz.

**Validation:** Sharona T. Levy.

**Writing – original draft:** Barak Garty.

**Writing – review & editing:** Gil Gambash, Sharona T. Levy, Guy Bar-Oz.

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
