## [Decision Letter · Decision Letter 0]

PONE-D-25-13772Wine economy in Byzantine Shivta (Negev, Israel): Exploring the role of runoff agriculture and droughts through Agent-Based ModelingPLOS ONE

Dear Dr. Garty,

Thank you for submitting your manuscript to PLOS ONE. After careful consideration, we feel that it has merit but does not fully meet PLOS ONE’s publication criteria as it currently stands. Therefore, we invite you to submit a revised version of the manuscript that addresses the points raised during the review process.

The reviewers were on a whole positive about this very interesting study, however they offer numerous comments that can improve the manuscript. Of particular importance is the clarification of the calculations of different data (such as labor, as shown in Table 2). Please consider these comments and return the corrected manuscript for further consideration.

We look forward to receiving your revised manuscript.

Kind regards,

Joe Uziel

Academic Editor

PLOS ONE

“G. B. O. : European Union European Research Council (ERC) Horizon 2020 Research and Innovation Program (Grant No. 101096539)”

“The authors wish to thank Yoav Avni, Yoav Borstein, Hendrick Bruins, Daniel Fuks, Noam Greenberg, Shauly Melitz, Efrat Morin, Moti Zohar for helping us with various aspects of this project and for contributing their knowledge and time. B.G. is grateful to Sir Maurice and Lady Irene Hatter Foundation for providing a research grant. The project received funding from the European Union European Research Council (ERC) Horizon 2020 Research and Innovation Program (Grant No. 101096539).”

“G. B. O. : European Union European Research Council (ERC) Horizon 2020 Research and Innovation Program (Grant No. 101096539)”

6. We note that Figures 1, 2 and Fig. E in your submission contain [map/satellite] images which may be copyrighted. All PLOS content is published under the Creative Commons Attribution License (CC BY 4.0), which means that the manuscript, images, and Supporting Information files will be freely available online, and any third party is permitted to access, download, copy, distribute, and use these materials in any way, even commercially, with proper attribution. For these reasons, we cannot publish previously copyrighted maps or satellite images created using proprietary data, such as Google software (Google Maps, Street View, and Earth). For more information, see our copyright guidelines: http://journals.plos.org/plosone/s/licenses-and-copyright.

a. You may seek permission from the original copyright holder of Figures 1, 2 and Fig. E to publish the content specifically under the CC BY 4.0 license. 

Reviewers' comments:

Reviewer's Responses to Questions

**Comments to the Author**

1. Is the manuscript technically sound, and do the data support the conclusions?

Reviewer #1: Yes

Reviewer #2: Yes

Reviewer #3: No

2. Has the statistical analysis been performed appropriately and rigorously? 

Reviewer #1: Yes

Reviewer #2: Yes

Reviewer #3: No

3. Have the authors made all data underlying the findings in their manuscript fully available?

Reviewer #1: Yes

Reviewer #2: Yes

Reviewer #3: No

4. Is the manuscript presented in an intelligible fashion and written in standard English?

Reviewer #1: Yes

Reviewer #2: Yes

Reviewer #3: Yes

5. Review Comments to the Author

Reviewer #1: 1) Page 3: “the thriving wine industry that

relied on a strong export market, which began in the late 4th to early 5th centuries and peaked in the

6th century before sharply declining in the mid-7th century CE [22�23]. This decline corresponds

with the periods of prosperity and downturn in Negev viticulture and wineries [14].”

Please note the decline in wine production and export from the Central Negev Highlands begins in the mid-6th century. By the mid-7th century, the export market did not just “decline” but had rather ceased altogether. The sentence on Page 3 is somewhat mitigated by the more correct details at the end of the paper on Page 24. Please make the proper correction on Page 3.

2) This reviewer is surprised that the authors made no references to recently published evidence from the nearby site and regional capital of Elusa. In general, there is a lack references to nearly any recently published data from sites other than Shivta itself, e.g., Elusa and Avdat. I suggest that this be rectified.

See: Heinzelmann, Michael – Schöne, Christian A. – Schröder, Arne – Wozniok, Diana –Jordan, Florian – Erickson-Gini, Tali – Kühn, Marlu – Langgut, Dafna – Lehnig, Sina. 2022. Elusa – from Nabatean Trading Post to Late Antique Desert Metropolis. Results of the 2015−2020 Seasons. Archäologischer Anzeiger 2022/1. DOI: https://doi.org/10.34780/di18-ld82

3) This reviewer was further surprised to see no mention of an important, relevant reference from Procopius of Gaza (450-526) concerning the condition of vines at Elusa during his lifetime.

See Mayerson P. 1985. The Wine and Vineyards of Gaza in the Byzantine Period. BASOR 257: 75‒80.

4) The map in Figure 1 is incomplete and other sites, particularly the capital of the region at Elusa, needs to be included. In addition, the site name, Nitzana, needs to be revised to: Nessana (as in the Nessana papyri).

5) The map in Figure 2 includes the name: Shivta city. Shivta was not a city but a large village / settlement. Please delete the word city.

6) The same for the table in Figure 3: please delete the word: city.

7) The resolution of Figure 4 is very poor. This reviewer was unable to read the key.

8) The name Zetan also appears as Zethan in Garty’s supplemental file. This and any other place names need to be spelled the same throughout the article and in the figures and supplementary files.

Reviewer #2: General Assessment

This study presents a well-structured and methodologically rigorous analysis of viticulture in Byzantine Shivta, integrating archaeological, climatic, and economic data with an innovative agent-based modeling (ABM) approach. The manuscript makes a valuable contribution to our understanding of agricultural resilience in marginal environments during Late Antiquity. The interdisciplinary methodology is particularly commendable, offering a quantitative framework for assessing the sustainability of ancient dryland viticulture.

While the manuscript is of high quality and merits publication, a few areas require minor revisions to enhance clarity, completeness, and methodological robustness. These revisions primarily concern historical context, data interpretation, and presentation of visual materials. Given the significance of the study and its solid foundation, I recommend acceptance with minor revisions.

Strengths of the Manuscript

1. Methodological Innovation: The use of an agent-based model (AGENTS) to simulate viticulture dynamics in the Negev Desert is a novel and commendable approach. The integration of NetLogo for environmental and economic simulations adds depth to the analysis.

2. Thorough Analysis of Runoff Agriculture: The discussion of Byzantine dryland farming techniques, including terraces, runoff harvesting, and water management strategies, is well-supported by archaeological and environmental data.

3. Consideration of Climatic Factors: The study provides a strong quantitative basis for assessing the impact of droughts and climatic variability on agricultural resilience in Shivta.

4. Economic Context: The exploration of viticulture as both an agricultural and economic enterprise is well-articulated, providing insights into the economic drivers behind Byzantine settlement sustainability.

Suggested Revisions

1. Incorporation of Additional Historical Context

o The manuscript does not consider the potential impact of the Plague of Justinian (541–549 CE) on the economic downturn of viticulture in Shivta. While recent studies suggest that the plague’s influence was not as catastrophic as once believed (see EISENBERG and MORDECHAI 2020 The Justinianic Plague and Global Pandemics: The Making of the Plague Concept), it remains an important factor to acknowledge when discussing economic resilience in Late Antiquity. A brief discussion of this context would strengthen the manuscript’s historical grounding.

o The correspondence of Procopius of Caesarea with Jerome references droughts and sandstorms in the fourth century that damaged vineyards and increased salinity in local wells (Mayerson 1983, The City of Elusa in the Literary Sources of the Fourth-Sixth Centuries. IEJ 32, 247-53 [252–3]). Including this historical source would complement the climatic data and reinforce the study’s conclusions on environmental challenges.

o Procopius also remarks on the high value of locally produced wine in Byzantine Shivta, which would further support the economic analysis presented in the manuscript.

2. Comparison with Modern and Pre-Modern Viticulture

o The manuscript compares Byzantine viticulture in Shivta with 17th-century Italian vineyards and contemporary rainfed viticulture in Spain. While these comparisons provide some useful context, they are problematic due to significant differences in climate, soil composition, available technology, and even grape varieties. Given these variables, the direct application of these case studies to Byzantine Shivta should be reconsidered or qualified with greater nuance.

3. Role of Pigeon Towers and Soil Fertility

o While the manuscript mentions pigeon towers, it does not discuss their role in soil enrichment through bird droppings, which could have significantly influenced vineyard productivity. Including an analysis of this factor would provide a more comprehensive picture of Byzantine agricultural practices.

4. Bibliographic and Editorial Refinements

o The bibliography, while extensive and well-researched, contains minor inconsistencies and typographical errors. A careful editorial review would enhance the manuscript’s overall professionalism and readability.

5. Figures and Data Presentation

o The necessity of Figure 3 should be reconsidered, as it does not appear to be critical to the manuscript’s argument. If it does not provide essential data, its removal could streamline the presentation.

Conclusion and Recommendation

The manuscript presents a strong and well-supported analysis of Byzantine viticulture in the Negev Desert. The methodological approach is innovative, and the findings provide valuable insights into ancient agricultural sustainability. While the study is highly commendable, addressing the minor revisions outlined above will further strengthen its contribution.

Final Recommendation: Accept with Minor Revisions.

I appreciate the opportunity to review this manuscript and look forward to seeing the revised version.

Reviewer #3: Comments were attached for the editor, and i hope that these will be shared with the authors.

Wearing my ancient historian hat before i considered the modeling, i approached the primary sources that jumped out at me, especially on page 13 which contains table 2: here, there were two issues. The first was that -- when i checked the text of citations -- Varro or Columella etc. did not say what was claimed. I lay out all these issues in the text shared with editors. The second problem was that -- besides the ancient primary sources -- there were a number of references to modern sources like *Negev Challenge of a Desert* or Duncan-Jones on Roman economy that lacked page numbers, and so i could not check the provided figures.

All of these values were critical for the modeling being used in the article, and thereby its conclusions: with disagreement or a lack of reproducibility, there is no other choice but "revise and resubmit."

I will add, however, that the modeling approach used here looks quite interesting! I have some concerns about how applicable it might be for other regions -- or indeed if the ancient primary sources here really have much validity in the ways that the authors want to use them, especially outside Italy. But it's a valuable attempt.

6. PLOS authors have the option to publish the peer review history of their article (what does this mean? ). If published, this will include your full peer review and any attached files.

**Do you want your identity to be public for this peer review?** For information about this choice, including consent withdrawal, please see our Privacy Policy .

Reviewer #1: No

Reviewer #2: No

Reviewer #3: No

---

## [Author Response · Author response to Decision Letter 1]

6 May 2025

Letter to PlosOne editor – response to comments

Dear editor, please find below our point-by-point response to the concerns and suggestions raised. We provide our responses after each comment in red. We thank the editorial team and all reviewers for the careful reading of the manuscript and the constructive comments provided. The suggestions and clarifications were helpful in improving the accuracy, structure, and contextual framing of the article. Below we outline the changes made in response to each reviewer comment. Where revisions were not made, we provide a brief explanation. We thank you for considering our work for publication in PLoS One. Respectfully yours, Barak Garty.

Subject: PLOS ONE Decision: Revision required [PONE-D-25-13772; PONE-D-25-13772R1]

Response: Done

Response: The AGENTS model code and supporting materials (data files and documentation) are publicly available on GitHub:

https://github.com/BarakGarty/AGENTS.

The model is also archived with a DOI on COMSES: https://doi.org/10.25937/b356-wh64.

“G. B. O. : European Union European Research Council (ERC) Horizon 2020 Research and Innovation Program (Grant No. 101096539)”

Response: "The funders had no role in study design, data collection and analysis, decision to publish, or preparation of the manuscript."

This was also added to revised cover letter that was upload to the revision submission.

“The authors wish to thank Yoav Avni, Yoav Borstein, Hendrick Bruins, Daniel Fuks, Noam Greenberg, Shauly Melitz, Efrat Morin, Moti Zohar for helping us with various aspects of this project and for contributing their knowledge and time. B.G. is grateful to Sir Maurice and Lady Irene Hatter Foundation for providing a research grant. The project received funding from the European Union European Research Council (ERC) Horizon 2020 Research and Innovation Program (Grant No. 101096539).”

“G. B. O. : European Union European Research Council (ERC) Horizon 2020 Research and Innovation Program (Grant No. 101096539)”

Response: Funding-related text was removed from the Acknowledgments section. All funding information now appears solely in the Funding Statement section, as per journal policy. In addition, our current Funding Statement “G. B. O.: European Union European Research Council (ERC) Horizon 2020 Research and Innovation Program (Grant No. 101096539)”

is sufficient.

Response: All data used in the research is now publicly available in the GitHub repository: https://github.com/BarakGarty/AGENTS. This includes a compressed file titled “Research results.rar,” which contains the Excel datasets derived from the AGENTS model output and used for the statistical analysis performed in SPSS.

6. We note that Figures 1, 2 and Fig. E in your submission contain [map/satellite] images which may be copyrighted. All PLOS content is published under the Creative Commons Attribution License (CC BY 4.0), which means that the manuscript, images, and Supporting Information files will be freely available online, and any third party is permitted to access, download, copy, distribute, and use these materials in any way, even commercially, with proper attribution. For these reasons, we cannot publish previously copyrighted maps or satellite images created using proprietary data, such as Google software (Google Maps, Street View, and Earth). For more information, see our copyright guidelines: http://journals.plos.org/plosone/s/licenses-and-copyright.

a. You may seek permission from the original copyright holder of Figures 1, 2 and Fig. E to publish the content specifically under the CC BY 4.0 license.

Response: Figures 1, 2 and Fig. E were created using ArcMAP by ESRI with our own layers except the basemap which is Esri ArcMAP software component.

Esri permits the use of its basemaps in publications, including scientific articles, provided proper attribution is included, as outlined in Esri’s guidelines.

In accordance with these requirements, we have taken the following steps:

Figure Captions: An attribution statement has been added to the captions of all relevant figures: “Basemap source: Esri, Maxar, Earthstar Geographics, and the GIS User Community.”

Main Text Statement: A general statement was added at the end of the article (after the Supporting Information captions) clarifying that maps throughout the manuscript were created using ArcMap® (version 10.8.2) by Esri, incorporating the World Imagery basemap.

Reference List Entry: A citation for the basemap was added to the reference list:

Esri. “World Imagery” [basemap]. Scale not given. “World Imagery Map.” 2009. Available from:

https://www.arcgis.com/home/item.html?id=10df2279f9684e4a9f6a7f08febac2a9

We trust that these actions align with both Esri’s attribution policy and PLOS ONE’s CC BY 4.0 licensing requirements.

Response: Reference list was reviewed and all style issues found were corrected.

To academic editor (General changes): In response to reviewer #3's comments regarding the complexity of Table 2, we decided to add a new supplemental file (S8) that explains the values presented in the table, the calculations used to derive them, and the specific input used in the AGENTS model for each agricultural activity. Accordingly, all other supplemental files were renumbered based on the order in which they are cited in the manuscript. Page numbers were also added to the source table, addressing Reviewer #3’s comment regarding precise source referencing.

Reviewer's Responses to Questions

Comments to the Author

1. Is the manuscript technically sound, and do the data support the conclusions?

Reviewer #1: Yes

Reviewer #2: Yes

Reviewer #3: No

Response: Reviewer 3’s concerns regarding the manuscript’s technical soundness, specifically the cited sources in Table 2 and the calculations used to derive the values in that table, have been addressed. We have thoroughly reviewed Table 2 and sources that did not provide accurate data were removed and we added calculations are now documented in the newly added Supplemental File S8.

2. Has the statistical analysis been performed appropriately and rigorously?

Reviewer #1: Yes

Reviewer #2: Yes

Reviewer #3: No

Response: Reviewer #3 responded “No” to the question regarding whether the statistical analysis was performed appropriately and rigorously, but did not provide specific comments elaborating on this concern. As such, we are not entirely certain what aspect of the analysis was in question.

That said, we revisited the relevant parts of the manuscript to ensure clarity and accuracy. Table 2, which includes key model outputs, was reviewed and revised after identifying some values that were either unclear or based on misleading sources. A new supplemental file (S8) was also added, providing a step-by-step explanation of how each value in Table 2 was calculated.

All model output used in the study is fully available in the GitHub repository, as indicated in the Data Availability section.

3. Have the authors made all data underlying the findings in their manuscript fully available?

Reviewer #1: Yes

Reviewer #2: Yes

Reviewer #3: No

Response: Data is now available on GitHub including the model code and model output used in the SPSS analysis.

4. Is the manuscript presented in an intelligible fashion and written in standard English?

Reviewer #1: Yes

Reviewer #2: Yes

Reviewer #3: Yes

5. Review Comments to the Author

Reviewer #1: 1) Page 3: “the thriving wine industry that relied on a strong export market, which began in the late 4th to early 5th centuries and peaked in the 6th century before sharply declining in the mid-7th century CE [22�23]. This decline corresponds with the periods of prosperity and downturn in Negev viticulture and wineries [14].”

Please note the decline in wine production and export from the Central Negev Highlands begins in the mid-6th century. By the mid-7th century, the export market did not just “decline” but had rather ceased altogether. The sentence on Page 3 is somewhat mitigated by the more correct details at the end of the paper on Page 24. Please make the proper correction on Page 3.

Response: The sentence on page 3 (lines 68�69) was revised to more accurately reflect the historical timeline, stating that the decline began in the mid-6th century CE. References were also update.

2) This reviewer is surprised that the authors made no references to recently published evidence from the nearby site an

---

## [Decision Letter · Decision Letter 1]

Wine economy in Byzantine Shivta (Negev, Israel): Exploring the role of runoff agriculture and droughts through Agent-Based Modeling

PONE-D-25-13772R1

Dear Dr. Garty,

We’re pleased to inform you that your manuscript has been judged scientifically suitable for publication and will be formally accepted for publication once it meets all outstanding technical requirements.

Kind regards,

Joe Uziel

Academic Editor

PLOS ONE

Additional Editor Comments (optional):

Reviewers' comments:

Reviewer's Responses to Questions

**Comments to the Author**

1. If the authors have adequately addressed your comments raised in a previous round of review and you feel that this manuscript is now acceptable for publication, you may indicate that here to bypass the “Comments to the Author” section, enter your conflict of interest statement in the “Confidential to Editor” section, and submit your "Accept" recommendation.

Reviewer #3: All comments have been addressed

2. Is the manuscript technically sound, and do the data support the conclusions?

Reviewer #3: Yes

3. Has the statistical analysis been performed appropriately and rigorously? 

Reviewer #3: Yes

4. Have the authors made all data underlying the findings in their manuscript fully available?

Reviewer #3: Yes

5. Is the manuscript presented in an intelligible fashion and written in standard English?

Reviewer #3: Yes

6. Review Comments to the Author

Reviewer #3: My concerns from the last review have been addressed: i appreciate the authors' efforts to check, revise, and clarify labor calculations. My recommendation is to accept.

7. PLOS authors have the option to publish the peer review history of their article (what does this mean? ). If published, this will include your full peer review and any attached files.

**Do you want your identity to be public for this peer review?** For information about this choice, including consent withdrawal, please see our Privacy Policy .

Reviewer #3: No

---

## [Editor Report · Acceptance letter]

PONE-D-25-13772R1

PLOS ONE

Dear Dr. Garty,

I'm pleased to inform you that your manuscript has been deemed suitable for publication in PLOS ONE. Congratulations! Your manuscript is now being handed over to our production team.

Kind regards,

on behalf of

Dr. Joe Uziel

Academic Editor

PLOS ONE